# Escaping fronts in local quenches of a confining spin chain

Anna Krasznai[1,2] and Gábor Takács[1,2,3]

**1** Department of Theoretical Physics, Institute of Physics,
Budapest University of Technology and Economics,
Műegyetem rkp. 3., H-1111 Budapest, Hungary
**2** BME-MTA Statistical Field Theory 'Lendület' Research Group,
Budapest University of Technology and Economics,
Műegyetem rkp. 3., H-1111 Budapest, Hungary
**3** MTA-BME Quantum Dynamics and Correlations Research Group,
Budapest University of Technology and Economics,
Műegyetem rkp. 3., H-1111 Budapest, Hungary

## Abstract

We consider local quenches from initial states generated by a single spin-flip in either the true or the false vacuum state of the confining quantum Ising spin chain in the ferromagnetic regime. Contrary to global quenches, where the light-cone behaviour is strongly suppressed, we find a significant light-cone signal propagating with a nonzero velocity besides the expected localised oscillating component. Combining an analytic representation of the initial state with a numerical description of the relevant excitations using the two-fermion approximation, we can construct the spectrum of post-quench excitations and their overlaps with the initial state, identifying the underlying mechanism. For confining quenches built upon the true vacuum, the propagating signal consists of superpositions of left and right-moving mesons escaping confinement. In contrast, for anti-confining quenches built upon the false vacuum it is composed of superpositions of left and right-moving bubbles which escape Wannier-Stark localisation.

 Check for updates

# 1   Introduction

Confinement is a physical phenomenon featuring prominently in the theory of strong interactions [1], explaining the absence of quarks and gluons as free particles. Besides the context of particle physics, confining forces also appear in condensed matter systems, with the most straightforward example provided by the McCoy-Wu scenario [2] in the ferromagnetic phase of the Ising field theory, which describes the scaling limit of the quantum Ising spin chain.

It was shown in [3] that the non-equilibrium dynamics of the quantum Ising spin chain is radically altered by confinement. In this context, non-equilibrium time evolution was initiated by a quantum quench, i.e., a sudden change in the Hamiltonian [4, 5]. This protocol is of paradigmatic relevance, and today, it is routinely engineered in experiments on closed quantum systems [6–13]. In a large class of such systems, including various spin chains with short-range interactions or interacting bosonic or fermionic systems, quasi-particles have an upper bound on their velocity [14], leading typically to a distinctive light-cone pattern in time-dependent correlation functions [4, 5, 15–20]. However, as shown in [3], the confining forces can suppress the light-cone spreading of correlations. Dynamical confinement and its effects have recently been confirmed in numerous systems [21–34].

Using the paradigmatic Ising chain in its confining regime, it was found that transport is also strongly suppressed in inhomogeneous situations, such as a quench starting from a domain wall state [22]. This effect is due to a combination of dynamical confinement and Wannier-Stark localisation, as demonstrated for the Ising spin chain in [26]. In contrast to the usual confinement, which is also present in the scaling field theory limit, Wannier-Stark localisation results from Bloch oscillations specific to condensed matter systems defined on a lattice [35]. Bloch oscillations were also shown to suppress the decay of the false vacuum under the standard scenario of bubble nucleation [36,37] by preventing the expansion of the nucleated bubbles of the true vacuum [38]. We note that bubble nucleation in the Ising spin chain was recently observed in numerical simulations of quantum quenches [39], confirming earlier theoretical predictions [40], with the method recently extended to quantum field theories [41, 42]. Approaching the decay of the false vacuum via quantum quenches in condensed matter systems opened the way to the quantum simulation of the false vacuum decay [42–46], which is also essential for high energy physics and cosmology [47–49]. Furthermore, a recent work [50] even raised the fascinating possibility of diagnosing a long-lived false vacuum using quantum quenches without precipitating the eventual vacuum decay.

Here, we study the dynamics of the ferromagnetic Ising spin chain with local quenches starting from initial states obtained by flipping a single spin. An advantage of these protocols is that dynamical confinement can be entirely separated from Wannier-Stark localisation, the former appearing in confining quenches where the initial magnetisation is aligned with the post-quench longitudinal field, while the latter arising when they are opposite. These correspond to the initial states obtained by a local change in the post-quench Hamiltonian's true/false vacuum state. In contrast to quenches starting from a domain wall, we observe very prominent escaping fronts tracing out a well-defined light cone in the time evolution of the magnetisation. We explain the origin of these escaping fronts using a combination of exact analytic and numerical approaches. This leads to the conclusion that the relevant states are equal-weight superpositions of left and right-moving excitations.

The outline of the present work is as follows. In Section 2, we briefly recall confinement in the Ising spin chain and describe the phenomenology of local quenches obtained by numerical simulation, while Section 3 discusses the meson and bubble spectrum in the two-fermion approximation. In Section 4, we consider confining quenches, corresponding to the case when the post-quench magnetic field is aligned with the pre-quench magnetisation, by developing a semi-analytic approach to compute the quench overlaps, i.e., the overlaps of meson excitations with the initial state. Section 5 focuses on anti-confining quenches, i.e., those when the post-quench magnetic field is opposite to the pre-quench magnetisation. Our conclusions are presented in Section 6. To keep the flow of the argument uninterrupted, we relegated some of the details to appendices, with Appendix A recalling the necessary ingredients of the exact solution of the purely transverse chain, Appendix B illustrates the wave function matching discussed in the main text, while Appendix C presents some examples of other local quenches intended for comparison.

## 2 Local quenches in the confining Ising spin chain

We consider the Ising spin chain with both transverse and longitudinal fields given by the Hamiltonian

$$H = -J \sum_{j=1}^{L} \left( \hat{\sigma}_j^x \hat{\sigma}_{j+1}^x + h_t \hat{\sigma}_j^z + h_l \hat{\sigma}_j^x \right), \tag{1}$$

where $L$ is the length of the chain, while $h_t$ and $h_l$ are the transverse/longitudinal magnetic fields, respectively. In writing (1) we implicitly used periodic boundary conditions $\sigma_{L+1}^a \equiv \sigma_1^a$. However, for the phenomena considered here, the boundary condition is irrelevant, and we choose it to be convenient for the given calculation, as specified later. The lattice spacing is taken to be $a = 1$, while $J$ specifies the units of energy and time.

For a vanishing longitudinal field, the model is exactly solvable. To keep our exposition self-contained, a brief overview of the exact solution of the chain for periodic boundary conditions is given in Appendix A. In the ferromagnetic phase $h_t < 1$, the spectrum consists of kink excitations with a dispersion relation

$$\varepsilon_q = 2J \sqrt{1 + h_t^2 - 2h_t \cos q}, \tag{2}$$

with the momentum $q$ running over the Brillouin zone $-\pi < q \leq \pi$. From here on, we also set $J = 1$ to specify our units of energy and time. The order parameter has a non-vanishing expectation value $\langle \hat{\sigma}_i^x \rangle = m$ with

$$m = \pm \left( 1 - h_t^2 \right)^{1/8}, \tag{3}$$

in the two ground states $|\pm\rangle_{h_t, h_l=0}$ which are degenerate in the thermodynamic limit.

## 2.1 Confinement in the Ising spin chain

Switching on a longitudinal magnetic field $h_l$ parallel to the magnetisation $m$ realises the McCoy-Wu scenario [2], leading to a linear potential with positive string tension

$$\chi = 2Jh_l m \,, \tag{4}$$

between kinks, which confines them into meson excitations. In the scaling limit, the disappearance of kinks from the spectrum can also be explained in field-theoretic terms based upon the mutual non-locality of the order parameter and the interpolating field of the kinks [51].

Choosing for definiteness $m > 0$, the true ground state of the system is a perturbation of the one with positive magnetisation $|+\rangle_{h_t,h_l=0}$, while $|-\rangle_{h_t,h_l=0}$ becomes a metastable state, a.k.a. the false vacuum. In the following, we stay within the scenario of strong confinement, which corresponds to a transverse field $h_t$ far below its critical value 1 and consider values of the longitudinal field $h_l$ of order 0.1.

When the system is not too close to the critical point, the meson excitations are almost purely composed of two-kink states of the $h_l = 0$ system and (for infinite volume) can be described in the so-called two-kink approximation with the effective Hamiltonian [52]

$$\hat{H}_{2-\text{fermion}} = \varepsilon(\hat{k}_1) + \varepsilon(\hat{k}_2) + \chi \, |\hat{x}_1 - \hat{x}_2| \,, \tag{5}$$

where $\hat{k}_{1,2}$ and $\hat{x}_{1,2}$ are the canonical momentum and coordinate operators for the kinks. The two-kink state dominance of the meson states was explicitly demonstrated in the field-theoretic limit [53], and it is related to the suppression of kink number changing processes [54].

The spectrum of this Hamiltonian can be computed either by a numerical solution of the Schrödinger equation or using a simple semi-classical approximation [52]. We avoid using values that are too large for $h_l$ to ensure that even the most spatially localised meson excitations have support over several lattice sites, which makes it possible to neglect the short-range deviations from the linear potential. These were computed in [52] and result from the kinks not being exactly point-like: The interpolation between the two vacua takes place over a crossover region of finite length.

To realise a confining quench, one starts from a ground state polarised parallel to the longitudinal field in the post-quench Hamiltonian. As shown in [3], confining forces radically alter the time evolution after a global quantum quench, suppressing the characteristic light-cone spreading of correlations. Note that due to the translational invariance of the initial state after a global quench, only meson states with zero total momentum eigenvalue can contribute; therefore, the one-meson states are localised. Nevertheless, string breaking leads to multi-meson excitations (with zero total momentum), producing fronts escaping confinement. However, this effect is usually strongly suppressed: in the quenches within the ferromagnetic case considered in the original work [3], the "escaping fronts" observed in the two-point correlation functions were smaller by several orders of magnitude compared to the localised component. The reason behind this suppression is well-understood [27, 55, 56]: string breaking is analogous to pair creation in a homogeneous electric field known as the Schwinger effect [57], with the role of the electric field played by the string tension (4). For small values of $h_t$ and $h_l$, the rate of string breaking is approximately given by [26]

$$Jh_t \exp\left(-\mathcal{C}\frac{1}{\sqrt{h_l^2 + h_t^2}}\right), \tag{6}$$

where $\mathcal{C}$ is a coefficient of order one. The anomalously slow thermalisation dynamics exhibited by confining global quenches can also be understood in terms of multiple stages with well-separated time scales, with the system first relaxing towards a prethermal state [58].

We note that when the polarisation of the initial state is opposite to the longitudinal field present after the quench, the time evolution is again localised in the global case; however, in this case, the mechanism behind this effect is Wannier-Stark localisation due to Bloch oscillations [26]. This corresponds to starting the dynamics from a false vacuum state, whose decay is prevented by Bloch oscillations [38]. Contrary to confinement, which also occurs in the continuum limit [53], Wannier-Stark localisation is a lattice effect resulting from the dispersion relation's periodicity over the Brillouin zone.

## 2.2 Phenomenology of local quenches

The first quench protocol we consider is the *confining quench* starting from the initial state

$$|\Psi(0)\rangle = \sigma_{L/2}^z |+\rangle_{h_t,h_l} \,, \tag{7}$$

where $|+\rangle_{h_t,h_l}$ is the positively polarised ground state of the post-quench Hamiltonian (1) with some fixed values of $0 < h_t < 1$ and $h_l > 0$. Since this only differs from the post-quench true vacuum state by flipping the spin at the site $L/2$, the resulting quench is fully local without any global component, i.e. it starts with energy density localised at the sites $L/2$ and its two neighbours, making the evolution much cleaner.

The time evolution was numerically simulated by the time-evolving block decimation (TEBD) method [59] under the post-quench Hamiltonian

$$H = -J \sum_{j=1}^{L-1} \left( \hat{\sigma}_j^x \hat{\sigma}_{j+1}^x + \frac{1}{2} h_t \hat{\sigma}_j^z + \frac{1}{2} h_t \hat{\sigma}_{j+1}^z + \frac{1}{2} h_l \hat{\sigma}_j^x + \frac{1}{2} h_l \hat{\sigma}_{j+1}^x \right), \tag{8}$$

using first-order Trotter-Suzuki decomposition with the typical total chain length $L = 80$ and $J = 1$. The Hamiltonian (8) only differs from (1) in the boundary conditions; however, this is irrelevant in our discussions since the simulation time considered here is not long enough for quasi-particles to travel the half-length of the chain. Consequently, the results of our subsequent analysis carry over to the thermodynamic limit $L \to \infty$ as well.

As a first step, the system's ground state was found by applying the TEBD method with imaginary time evolution and was validated by comparing it to the exact diagonalisation results. We then applied the spin flip and simulated the real-time evolution of the system. For the imaginary time evolution, we chose the time step $dt = 0.05$, the total number of steps $N = 4000$ and the maximum bond dimension $\chi_{\max} = 50$. For the real-time evolution, the parameters were set to $dt = 0.01$ and $\chi_{\max} = 300$. The Schmidt values were truncated at $10^{-8}$ for both imaginary and real-time evolutions. We verified that all results were robust under reasonable variations of these parameters.

The resulting time evolution of the expectation value of the longitudinal magnetisation

$$M(j,t) = \langle \Psi(t)|\sigma_j^x|\Psi(t)\rangle \,, \qquad |\Psi(t)\rangle = e^{-iHt} |\Psi(0)\rangle \,, \tag{9}$$

is shown in Fig. 1. Note that the evolution has a prominent localised component similar to the global quenches considered in [3]. This is expected since a single flipped spin is essentially a source of kink pairs, whose motion is governed by the Hamiltonian (5) with confining potential.

In a semi-classical approximation, one can use the classical version of the Hamiltonian (5) to compute the trajectory of a kink pair by introducing the centre of mass and relative coordinates

$$X = \frac{x_1 + x_2}{2} \,, \qquad x = x_2 - x_1 \,, \tag{10}$$

together with the associated momenta

$$K = k_1 + k_2 \,, \qquad k = \frac{k_2 - k_1}{2} \,. \tag{11}$$

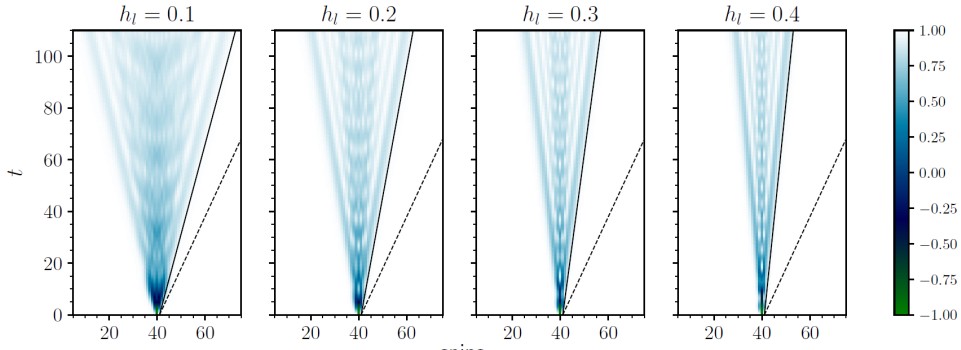

(a) The solid lines indicate the maximum meson velocities calculated from the corresponding meson spectrum as explained in Subsection 3.1, while the dashed lines indicate the maximum free kink velocity.

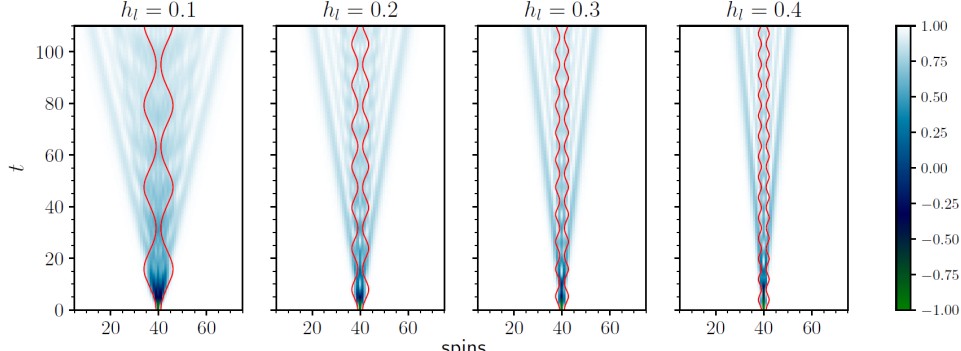

(b) The red curves indicate the corresponding quasi-classical trajectories of the quasi-particles with initial momenta $k_0 = \pm\pi$.

Figure 1: The front propagation in the longitudinal magnetisation $\langle \hat{\sigma}_j^x \rangle$ for $h_t = 0.25$ and $h_l \in \{0.1, 0.2, 0.3, 0.4\}$ respectively, starting from the initial state $|\Psi(0)\rangle = \sigma_{L/2}^z |+\rangle_{h_t, h_l}$.

For states with total momentum $K = 0$, we obtain the equation

$$\dot{x}(t) = 2\frac{\partial \epsilon(k)}{\partial k}, \qquad \dot{k}(t) = -\chi \, \text{sign}(x(t)). \tag{12}$$

Introducing $d(t) = |x(t)|/2$ as the position of the right kink relative to the midpoint, the equations of motion (12) can be solved exactly to give

$$d(t) = \left| -\frac{1}{\chi}(\epsilon(k_0 - \chi t) - \epsilon(k_0)) + d_0 \right|, \tag{13}$$

where $k_0 > 0$ gives the initial value of the kink momenta as $k_{1,2}(t = 0) = \pm k_0$, while $d_0 > 0$ gives their initial position as $x_{1,2}(t = 0) = \pm d_0 + X$, with their center located at $X$. In the local quenches considered here, the typical value of $d_0$ is expected to be of the order of the lattice spacing. At the same time, the enveloping curve of the localised component is given by the trajectory with $k_0 = \pi$. Fig. 1b shows the predicted enveloping curve with red lines choosing $d_0 = 1$.

Note that while the localised component qualitatively matches the predicted envelope, there is a component forming a prominent light cone which lies outside the red curves. In contrast to the results for global quenches [3], the escaping fronts in local quenches are of the same order of magnitude as the localised component. Similar dynamics was observed in [31];

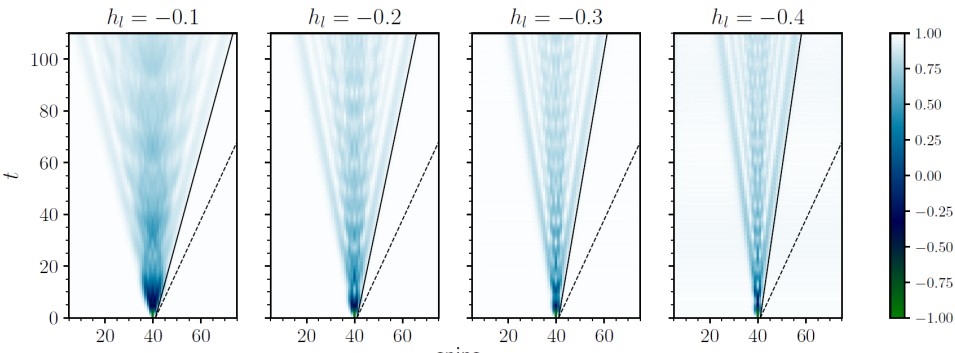

(a) The solid lines indicate the maximum bubble velocities calculated from the corresponding bubble spectrum as explained in Subsection 3.2, while the dashed lines indicate the maximum free kink velocity.

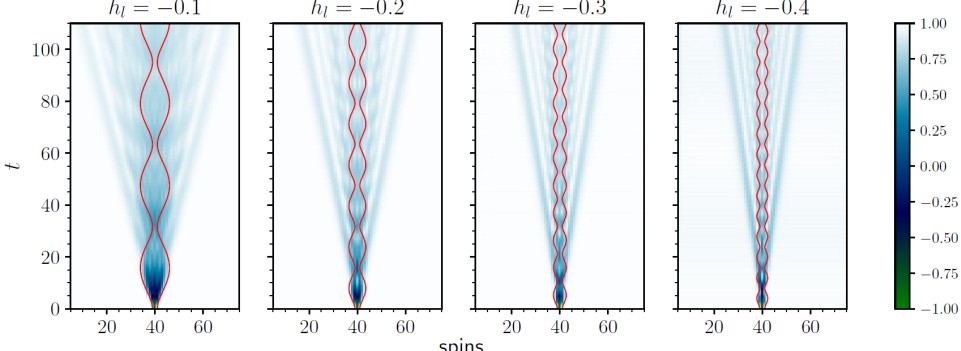

(b) The red curves indicate the corresponding quasi-classical trajectories of the kink pair with initial momentum $k_0 = 0$.

Figure 2: The front propagation in the longitudinal magnetisation $\langle \hat{\sigma}_j^x \rangle$ for $h_t = 0.25$ and $h_l \in \{-0.1, -0.2, -0.3, -0.4\}$ respectively, starting from the initial state $|\overline{\Psi}(0)\rangle = \sigma_{L/2}^z |+\rangle_{h_t, h_l=0}$.

however, the transverse field applied there was much larger ($h_t = 0.5$), much closer to the critical point. They also considered a different observable: The probability dynamics of kinks instead of the magnetisation which is considered here. It is striking to see such a prominent light-cone signal in the strongly confining regime where the light-cone in global quenches was observed to be very strongly suppressed [3]. Our primary interest is to identify the mechanism underlying their presence.

The second quench protocol is the *anti-confining quench* starting from the initial state

$$|\overline{\Psi}(0)\rangle = \sigma_{L/2}^z |+\rangle_{h_t, h_l=0} \, , \tag{14}$$

where $|+\rangle_{h_t, h_l=0}$ is the positively polarised ground state of the pure transverse Hamiltonian. In this case, we take the post-quench longitudinal field $h_l$ to be negative, and so the initial state corresponds to a metastable (false vacuum) state for the Hamiltonian (1). The general settings of the TEBD simulations for these quenches were the same as in the confining case. Note that such quenches are not purely local, i.e. they necessarily involve a global component; however, it turns out that this component is so small that it can be neglected in our subsequent considerations.

Despite the repulsive force between the kinks characteristic for this case, Bloch oscillations lead to Wannier-Stark localisation of the dynamics [26, 38], with the resulting evolution displayed in Fig. 2. While it looks superficially similar to the case of confining quenches il-

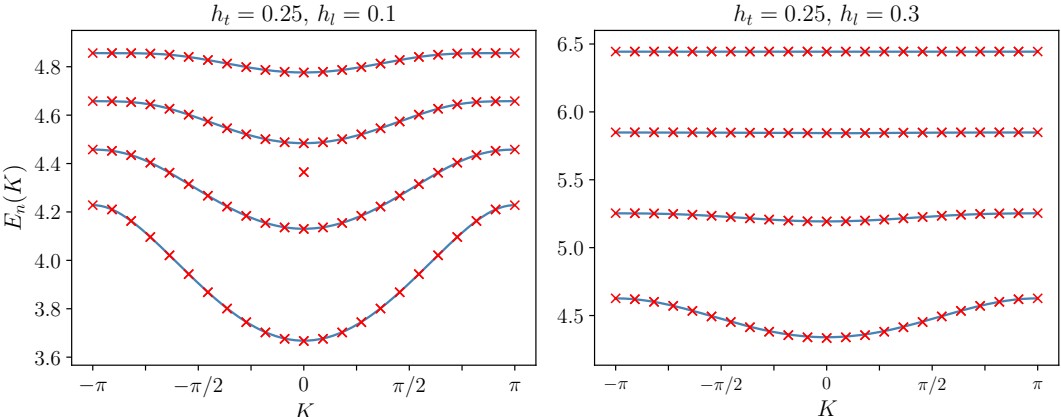

Figure 3: The lowest lying energy levels of the Schrödinger equation (16) for transverse field $h_t = 0.25$, and longitudinal fields $h_l = 0.1$ and $h_l = 0.3$ as a function of the momentum of the centre of mass $K$, shown by blue curves. The red crosses correspond to the energy eigenvalues calculated via exact diagonalisation for a chain of length $L = 22$. The energy of the real ground state is subtracted from each finite chain energy eigenvalue. The isolated red cross is the false vacuum state in the finite system with energy $\chi L$ relative to the true vacuum ground state.

lustrated in Fig. 1, we reveal significant differences in the underlying mechanism behind the escaping fronts.

# 3 The meson/bubble spectrum

In this Section, we consider the quasi-particle spectrum relevant for confining dynamics, which is composed of two-kink bound states known as mesons, as well as the excitations governing the anti-confining quenches which are nothing else than bubbles whose nucleation is the mechanism behind the decay of the false vacuum [36]. We use the two-fermion approximation of [52] to get a quantitative description whose accuracy can be cross-checked using exact diagonalisation.

## 3.1 Confinement: Mesons

As discussed in Subsection 2.1, the meson spectrum can be well approximated using the simple two-fermion Hamiltonian (5). Due to translational invariance, it is possible to pass to the centre-of-mass frame where the eigenvalue of the total momentum $\hat{k}_1 + \hat{k}_2$ is fixed to $K$. Then, the Hamiltonian for the relative motion takes the form

$$\hat{H} = \omega(\hat{k}; K) + \chi |\hat{x}|, \tag{15}$$

where $\omega(\hat{k}; K) = \varepsilon(K/2 + \hat{k}) + \varepsilon(K/2 - \hat{k})$. From now on, we will omit the hats from the operators. Considering an infinite chain with position $x$ taking values in $\mathbb{Z}$, the Schrödinger equation can be written as

$$H\psi_{n,K}(x) = \sum_{x'} H(x, x'; K)\psi_{n,K}(x') = E_n(K)\psi_{n,K}(x), \tag{16}$$

where the fermionic statistics implies that the wave functions are odd

$$\psi_{n,K}(x) = -\psi_{n,K}(-x), \tag{17}$$

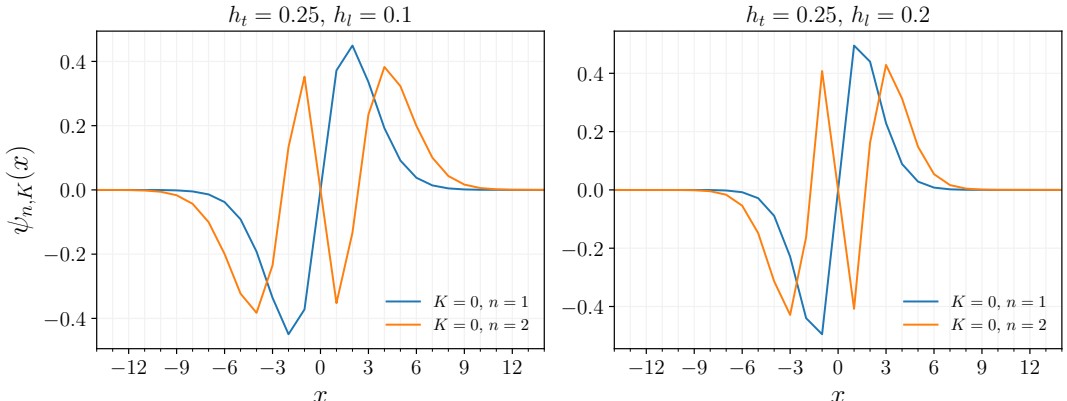

Figure 4: The meson wave functions $\psi_{n,K}(x)$ for the parameters $h_t = 0.25$, $h_l = 0.1$, and $h_l = 0.2$. The state with $K = 0$, $n = 1$ is shown in blue, while the $K = 0$, $n = 2$ case is indicated by a continuous orange curve obtained by connecting the discrete points where the wave function is defined.

and

$$H(x, x'; K) = H_0(x - x', K) + \chi |x| \delta_{x,x'},$$
$$H_0(x, K) = \int_{-\pi}^{\pi} \frac{dk}{2\pi} \{\varepsilon(K/2 + k) + \varepsilon(K/2 - k)\} e^{-ikx}. \tag{18}$$

Note that due to the (unit) lattice spacing, the momentum variables run over the Brillouin zone $(-\pi, \pi]$. It turns out that the kinetic term is localised around $x = 0$, and in practice, $H_0(x - x', K)$ can be considered as a narrow band matrix. In addition, the meson wave functions are themselves localised due to the confining potential, so an excellent approximation can be obtained by considering, at most, a few hundred sites instead of a chain of infinite length. In the calculations below, we use $-100 \leq x \leq 100$ and obtain the meson spectrum numerically from the finite matrix representing the Hamiltonian (16).

The $n$th level of the above Schrödinger problem corresponds to a meson one-particle state $|M_n(K)\rangle$ with momentum $K$ and species number $n$, with $E_n(K)$ giving its energy relative to the ground state in the two-fermion approximation, which we order to increase with $n$. For any value of $K$, the meson spectrum is bounded from below and unbounded from above.

In Fig. 3, we show the lowest four numerically obtained meson levels $E_n(K)$ with blue lines as a function of the momentum $K$ for the value of transversal field $h_t = 0.25$, with two values $h_l = 0.1$ and $h_l = 0.3$ for the longitudinal field. For comparison, we also display the spectrum obtained from the exact diagonalisation of the total spin chain Hamiltonian (1) with length $L = 22$ and the ground state value subtracted, which is shown by red crosses in Fig. 3. Note the excellent agreement confirming the validity of the two-fermion approximation.

The phase of the wave functions themselves can be chosen so that $\psi_{n,K}(x)$ is real. In Figure 4, we display some wave functions $\psi_{n,K}(x)$ to demonstrate their localisation. The spatial extension of the states decreases with $h_l$ but increases as the level index $n$ grows. Since the two-fermion approximation treats the kinks as point-like objects, it is only reliable when the extension of the meson wave function is substantially larger than the spatial size of the kinks. This is safely satisfied for the field values considered here (by at least an order of magnitude); nevertheless, as $h_l$ grows, the approximation gradually loses its validity, as discussed in Subsection 2.1.

Indeed, by implementing the Bogolyubov fermion operators (A.12, A.14) in the exact diagonalisation numerically, it can be verified that for the values of the transverse and longitudinal fields considered here, the partial norm squared of the meson eigenvectors obtained from the

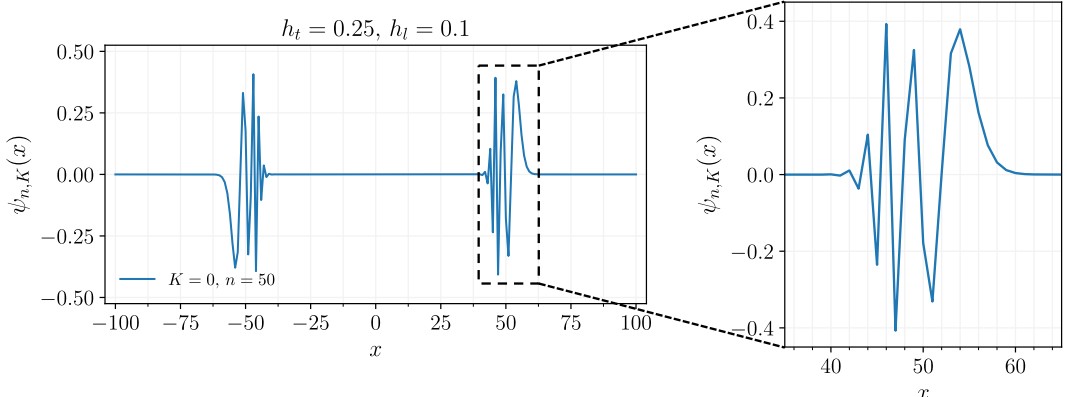

Figure 5: Left panel: The real part of the meson wave function $\psi_{n,K}(x)$ for the parameters $h_t = 0.25$, $h_l = 0.1$ and for $n = 50$. Note that the kinks are localised in a much smaller space than the distance between them, making the meson collisionless. Right panel: wave function for a single kink subject to Wannier-Stark localisation and undergoing Bloch oscillations.

spin chain outside the two-fermion subspace is at most of order $10^{-3}$, which explains the excellent accuracy of the two-fermion approximation.[1] Additionally, direct studies of scattering reveal that meson number changing amplitudes are greatly suppressed [54], which is also related to the suppression of string breaking discussed in Subsection 2.1.

It is apparent from Fig. 3 that for large species numbers $n$, the dispersion relation of mesons is very flat. The underlying mechanism is illustrated in Fig. 5: The two kinks are localised so far apart that they undergo separate Bloch oscillations without ever bumping into each other. This configuration can be called a collisionless meson and can be described simply in a semi-classical picture. The momenta of the two kinks satisfy

$$k_{1,2}(t) = q_{1,2} \mp \chi t, \tag{19}$$

with $q_{1,2}$ denoting their values at the reference time $t = 0$. Note that the total momentum $K = k_1 + k_2$ of the meson state is constant as expected. Using the velocity relation

$$\dot{x}_{1,2}(t) = \left.\frac{\partial \epsilon(k)}{\partial k}\right|_{k=k_{1,2}}, \tag{20}$$

the positions of the two kinks are given by

$$x_{1,2}(t) = \mp\frac{1}{\chi}\left(\epsilon(k_0 \mp \chi t) - \epsilon(k_0)\right) + x_{1,2}(0), \tag{21}$$

which is localised in a region of diameter

$$d_{\mathrm{WS}} = \frac{\epsilon(\pi) - \epsilon(0)}{\chi} = \frac{2h_t}{\chi}, \tag{22}$$

where $d_{\mathrm{WS}}$ denotes the Wannier-Stark localisation length. The corresponding mesons are collisionless for $|x_1(0) - x_2(0)| \gg d_{\mathrm{WS}}$, and their energy is given by the sum of the energies of the two localised kinks (which is independent of the total momentum) and the contribution

---

[1]For a similar result in the scaling field theory see Ref. [53].

from the stretch of false vacuum between them. This can be understood by setting up a semi-classical quantisation scheme in momentum space following [52], with the main distinction that the wave function does not have a turning point. Indeed, writing the wave function as

$$\psi(x) = \int_{-\pi}^{\pi} \frac{dk}{2\pi} \psi(k) e^{-ikx}, \tag{23}$$

the Schrödinger equation (16) can be written in momentum space as

$$(\epsilon(K/2+k) + \epsilon(K/2-k) - E)\psi(k) - i\chi\,\partial_k\psi(k) = 0, \tag{24}$$

which can be solved explicitly with the properly normalised result given by

$$\psi(k) = \exp\left\{\frac{i}{\chi}\left[Ek - \int_0^k dk'\left(\epsilon(K/2+k') + \epsilon(K/2-k')\right)\right]\right\}, \tag{25}$$

up to a phase factor. For the collisionless case, imposing periodicity of the wave function in the Brillouin zone as $\psi(k=\pi) = \psi(k=-\pi)$ leads to a Bohr-Sommerfeld quantisation condition of the form

$$2\pi E_n(K) - \int_{-\pi}^{\pi} dk\,[\epsilon(K/2+k) + \epsilon(K/2-k)] = 2\pi n\chi. \tag{26}$$

Due to the periodicity of $\epsilon(k)$ the integral term does not depend on $K$, leading to energy levels independent of $K$:

$$E_n(K) = n\chi + \frac{1}{\pi}\int_{-\pi}^{\pi} dk\,\epsilon(k), \tag{27}$$

which numerically matches the large-$n$ meson levels obtained from the Schrödinger equation (16) with a very high precision. The corresponding wave functions are given by antisymmetrising the solution (25):

$$\psi_{n,K}(k) = \frac{1}{\sqrt{2}} e^{ikn} \exp\left\{\frac{i}{\chi}\left[\frac{k}{\pi}\int_{-\pi}^{\pi} dk'\epsilon(k') - \int_0^k dk'\left(\epsilon(K/2+k') + \epsilon(K/2-k')\right)\right]\right\}$$
$$- \{k \to -k\}. \tag{28}$$

This can be checked to match numerically the wave function obtained by the explicit solution of (16) for large $n$. For the case $h_t = 0.25$ and $h_l = 0.1$, the agreement is already excellent for $n = 6$ and rapidly improves for larger $n$.

From the dispersion relation, the group velocity of mesons can be computed as

$$v_n(K) = \frac{\partial E_n(K)}{\partial K}. \tag{29}$$

It is clear from Fig. 3 that these functions have a maximum value inside the Brillouin zone. The existence of this maximum is guaranteed by the Lieb-Robinson theorem [14], and its value depends on the species $n$. For collisionless mesons (large $n$), the independence of their energy from their momentum $K$ implies that these mesons are stationary irrespective of the value of $K$, which is nicely explained by the semi-classical description. In addition, the velocities monotonously decrease with $n$; hence, the light-cone bounding the escaping fronts in Fig. 1a corresponds to the maximum velocity of the lightest meson species $M_1$.

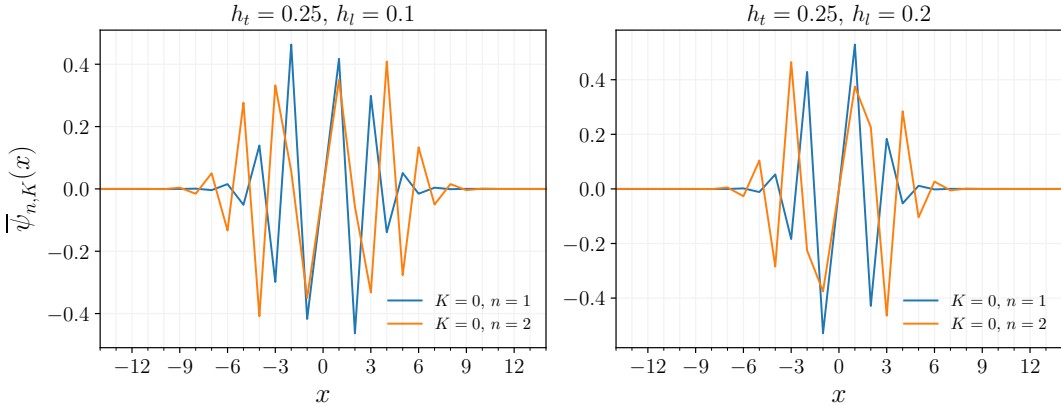

Figure 6: The (collisional) bubble wave functions $\overline{\psi}_{n,K}(x)$ for the parameters $h_t = 0.25$, $h_l = 0.1$, and $h_l = 0.2$. The state with $K = 0$, $n = 1$ is shown in blue, while a continuous orange curve indicates the $K = 0$, $n = 2$ case.

## 3.2 Anti-confinement: Bubbles

One can also investigate the spectrum built over the false vacuum in the two-fermion approximation. The two-kink states correspond to bubbles of the true vacuum nucleating under the well-known scenario of vacuum decay [36]. The relevant Hamiltonian differs from (15) in the sign of the string tension, resulting in the Schrödinger problem

$$\sum_{x'}\left[H_0(x - x', K) - |\chi||x|\right]\overline{\psi}_{n,K}(x') = \overline{E}_n(K)\overline{\psi}_{n,K}(x),$$

$$\overline{\psi}_{n,K}(x) = -\overline{\psi}_{n,K}(-x), \tag{30}$$

where we explicitly indicated the anti-confining nature of the force due to $\chi < 0$. The $n$th level of the above Schrödinger problem corresponds to a one-bubble state $|B_n(K)\rangle$ with momentum $K$ and species number $n$, with $\overline{E}_n(K)$ giving the energy of these states relative to the false vacuum state in the two-fermion approximation, which we order to decrease with $n$. For any value of $K$, the bubble spectrum is bounded from above and unbounded from below, just opposite to the behaviour of the mesons. These bubbles are created according to the standard scenario introduced in [36], and their nucleation rate was obtained in [40]. However, it was pointed out that the expansion of the nucleated bubbles and, therefore, the eventual decay of the false vacuum is suppressed by Bloch oscillations [38].

Examples of localised bubble wave functions $\overline{\psi}_{n,K}(x)$ obtained from (30) are shown in Figure 6. Similarly to mesons, their spatial extension decreases with growing $h_l$ but increases with the species index $n$. Comparison of Figs. 4 and 6 shows that the bubble wave functions exhibit much stronger oscillations than the meson ones. This corresponds to the fact that the meson momentum space wave functions included are centred around momentum $k = 0$ while the bubble wave functions presented there are localised around higher momentum values, as shown in Figs. 13 and 14. Similarly to mesons, bubbles with large $n$ are collisionless, i.e., composed of two kinks localised far away from each other, similarly to the situation shown for mesons in Fig. 5.

In a finite system of length $L$, the false vacuum has energy $|\chi|L$ relative to the true vacuum,[2] which must be considered when comparing the spectrum resulting from (30) to exact diagonalisation results. This comparison is shown in Fig. 7, where we chose the chain length $L = 14$ to display the single-meson spectrum together with the single-bubble spectrum while

---

[2]Neglecting finite size corrections decaying exponentially with $L$.

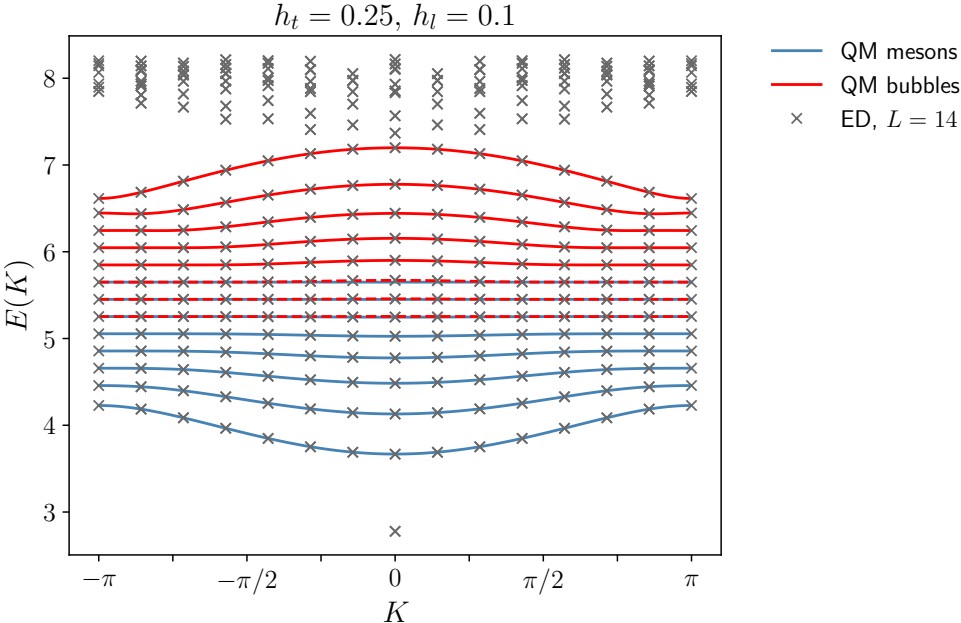

Figure 7: Low-energy spectrum of the TFIM with longitudinal field, relative to the energy of the true vacuum state with $h_t = 0.25$ and $h_l = 0.1$. The crosses show the first 295 energy eigenvalues calculated via exact diagonalisation for a chain of length $L = 14$, organised according to the total momentum $K$ of the corresponding state. The isolated cross is the false vacuum state, the blue curves display the meson energy levels $E_n(K)$ computed from the Schrödinger equation (16), while the red curves correspond to the bubble energy eigenvalues $\overline{E}_n(K)$ calculated from (30) shifted by the false vacuum energy $|\chi|L$. The middle of the spectrum shows that the two energy spectra agree in the collisionless limit as implied by Eq. (33).

keeping it separated from the higher excited states. Note that, on a periodic chain, large $n$ mesons are indistinguishable from large $n$ bubbles since the distinction between meson and bubbles is not absolute: Both are configurations divided by two kinks into two segments of true vs. false vacuum as shown in Fig. 8. This can be easily explained by considering large collisionless bubbles in semi-classical approximation, in which their quantisation can be obtained by flipping the sign of the string tension in (26):

$$2\pi\overline{E}_n(K) - \int_{-\pi}^{\pi} dk \left[ \epsilon(K/2 + k) + \epsilon(K/2 - k) \right] = -2\pi n|\chi|, \tag{31}$$

resulting in the solution

$$\overline{E}_n(K) = -n|\chi| + \frac{1}{\pi} \int_{-\pi}^{\pi} dk \, \epsilon(k), \tag{32}$$

a result which was also obtained in [40]. Adding the energy shift of the false vacuum, we see that it matches the spectrum of collisionless mesons by transforming the species index $n$ to $L - n$

$$|\chi|L + \overline{E}_n(K) = E_{L-n}(K), \tag{33}$$

which is exactly the correspondence between large bubbles and large mesons valid for finite length $L$, which was described above. Note that the energy relative to the false vacuum (32)

Figure 8: Illustrating the matching Eq. (33) between mesons and bubbles in a finite volume with periodic boundary conditions. The left panel shows an excitation that can be considered a meson, while the one on the right is more of a bubble. The green arrow shows the direction of the longitudinal field, the blue/red arrows depict spins polarised according to the true/false vacuum domains, and the green wiggly lines indicate the position of the kinks.

is positive for small bubbles and only becomes negative for bubbles larger than the critical size

$$n_* = \frac{1}{|\chi|} \int_{-\pi}^{\pi} \frac{dk}{\pi} \, \epsilon(k), \tag{34}$$

which grows monotonically with $h_t$, and is bounded from below by $2/|h_l|$. For the transverse and longitudinal fields considered here, the critical bubble size is always more than one order of magnitude larger than the lattice spacing, which plays an important role in suppressing vacuum decay as found in [38], a point to which we return later in the context of anti-confining quenches at the end of Section 5.

The group velocity of the bubbles can be evaluated the same way as for mesons:

$$\bar{v}_n(K) = \frac{\partial \overline{E}_n(K)}{\partial K}. \tag{35}$$

Note that in contrast to the mesons, the sign of the velocity is opposite to that of the momentum. However, these velocities still have finite maxima; the largest of these maximum velocities corresponds to the first bubble $B_1$, which is the one corresponding to the light-cone boundary of anti-confining quenches in Fig. 2a.

## 4 Escaping fronts in confining local quenches

In this Section, we consider the problem of escaping fronts in confining quenches and turn to anti-confining quenches in the next one.

Naively, the escaping fronts observed in the local quenches looked like as if a meson pair had traced them out. However, a simple energy estimate shows that this cannot be correct. The quench is generated by flipping a local spin, which does not inject enough energy into the system. For a crude estimate, note that the main contribution comes from the nearest neighbour interaction in (1), which can contribute at most $4J$ for a single spin-flip. The contribution by the longitudinal and transverse fields cannot be larger than $2J(h_l + h_t)$, so altogether, the injected energy is smaller than $(4 + 2h_l + 2h_t)J$. For all the quenches considered in Subsection 2.2, this is never enough to cover the energy of a meson pair. For example, taking $h_t = 0.25$ and $h_l = 0.1$, the injected energy is estimated from above by $5.2J$. At the same time, the 2-fermion approximation described in Section 3 yields $m_1 = 3.668J$ for the energy of the lightest meson with momentum zero, i.e. the meson gap. Therefore, the contribution of states with more than one meson is expected to be negligible.

The above considerations suggest that the explanation of the escaping fronts must be sought in terms of single-meson states. To reveal the nature of the states that contribute, we analyse the overlaps of the initial state with the post-quench meson states.

## 4.1  Initial state for $h_l = 0$

In the vanishing longitudinal field ($h_l = 0$) limit, the state (7) can be computed explicitly in terms of the fermions. According to (A.19), the system's ground state with positive magnetisation can be written as[3]

$$|+\rangle_{h_t, h_l=0} = \frac{1}{\sqrt{2}} \left( |0\rangle_{NS} + |0\rangle_R \right) . \tag{36}$$

In this approximation, the initial state (7) is

$$|\Psi(0)\rangle = \hat{\sigma}^z_{L/2} |+\rangle_{h_t, h_l=0} = \frac{1}{\sqrt{2}} \left( |\Psi(0)\rangle_{NS} + |\Psi(0)\rangle_R \right) , \tag{37}$$

where we separated the NS and R components. Using the Jordan-Wigner fermions (A.4), the operator $\hat{\sigma}^z_{L/2}$ can be expressed as

$$\hat{\sigma}^z_{L/2} = 1 - 2c^\dagger_{L/2} c_{L/2} = - \left( 1 - 2c_{L/2} c^\dagger_{L/2} \right) .$$

In the NS sector, substituting the Bogolyubov fermions using (A.12) gives

$$
\begin{aligned}
c_{L/2} &= \frac{1}{\sqrt{L}} \sum_{k \in NS} e^{-ikL/2} \left( u_k \eta_k + i v_k \eta^\dagger_{-k} \right) , \\
c^\dagger_{L/2} &= \frac{1}{\sqrt{L}} \sum_{k \in NS} e^{ikL/2} \left( u_k \eta^\dagger_k - i v_k \eta_{-k} \right) ,
\end{aligned}
\tag{38}
$$

from which one can obtain explicitly that

$$|\Psi(0)\rangle_{NS} = - \left\{ 1 - \frac{2}{L} \sum_{k \in NS} u_k^2 - \frac{2i}{L} \sum_{k_1, k_2 \in NS} e^{-i(k_1 - k_2)L/2} v_{k_1} u_{k_2} \eta^\dagger_{-k_1} \eta^\dagger_{k_2} \right\} |0\rangle_{NS} . \tag{39}$$

Similarly, in the Ramond sector, Eqs. (A.12, A.14) give

$$
\begin{aligned}
c_{L/2} &= \frac{1}{\sqrt{L}} \left\{ \eta^\dagger_0 + \sum_{\substack{p \in R \\ p \neq 0}} e^{-ipL/2} \left( u_p \eta_p + i v_p \eta^\dagger_{-p} \right) \right\} , \\
c^\dagger_{L/2} &= \frac{1}{\sqrt{L}} \left\{ \eta_0 + \sum_{\substack{p \in R \\ p \neq 0}} e^{ipL/2} \left( u_p \eta^\dagger_p - i v_p \eta_{-p} \right) \right\} .
\end{aligned}
\tag{40}
$$

Therefore, following the same argument as for the NS component, the R component can be written as

$$|\Psi(0)\rangle_R = - \left\{ 1 - \frac{2}{L} \sum_{\substack{p \in R \\ p \neq 0}} u_p^2 - \frac{2i}{L} \sum_{\substack{p_1, p_2 \in R \\ p_1 p_2 \neq 0}} e^{-i(p_1 - p_2)L/2} v_{p_1} u_{p_2} \eta^\dagger_{-p_1} \eta^\dagger_{p_2} - \frac{2}{L} \sum_{\substack{p \in R \\ p \neq 0}} e^{ipL/2} u_p \eta^\dagger_0 \eta^\dagger_p \right\} |0\rangle_R . \tag{41}$$

---

[3]Strictly speaking, the polarised states (A.19) are not eigenstates of the Hamiltonian for a finite chain length $L$. However, we neglect the corresponding exponentially small corrections in $L$ due to tunnelling effects throughout this work.

The required expansion of the state (37) is given by the sum of (39) and (41), while their difference gives the expansion of the state

$$\hat{\sigma}_{L/2}^{z} |-\rangle_{h_t, h_l=0} = \frac{1}{\sqrt{2}} \left( |\Psi(0)\rangle_{\mathrm{NS}} - |\Psi(0)\rangle_{\mathrm{R}} \right), \tag{42}$$

which is obtained by a spin-flip from the negatively polarised state

$$|-\rangle_{h_t, h_l=0} = \frac{1}{\sqrt{2}} \left( |0\rangle_{\mathrm{NS}} - |0\rangle_{\mathrm{R}} \right). \tag{43}$$

### 4.2 Matching the meson wave functions

To compute the meson overlaps, it is necessary to construct the meson wave functions in momentum space. On the one hand, on a finite chain of length $L$ and under the assumptions of the two-fermion approximation, the meson states can be expressed in terms of the Bogolyubov fermions as

$$|M_n(K)\rangle = \sum_{k \in \mathrm{NS}+K/2}' \widetilde{\psi}_{n,K}(k)_L \eta_{K/2-k}^{\dagger} \eta_{K/2+k}^{\dagger} |0\rangle_{\mathrm{NS}} + \sum_{k \in \mathrm{R}+K/2}' \widetilde{\psi}_{n,K}(k)_L \eta_{K/2-k}^{\dagger} \eta_{K/2+k}^{\dagger} |0\rangle_{\mathrm{R}}, \tag{44}$$

where $K$ is an integer multiple of $2\pi/L$ and the prime means that we avoid double counting (i.e. every pair of momenta $K/2 \pm k$ occurs only once). The notation $k \in \mathrm{NS/R} + K/2$ means that the momentum runs over the NS/R values shifted by $K/2$ (modulo $2\pi$), which ensures that $K/2 \pm k$ is in the NS/R sector, as required. These wave function amplitudes can be computed from exact diagonalisation as

$$\widetilde{\psi}_{n,K}(k)_L = {}_{\mathrm{NS/R}}\langle 0 | \eta_{K/2+k} \eta_{K/2-k} |M_n(K)\rangle, \tag{45}$$

where the vacuum state in the matrix element is chosen according to whether the fermion momenta $K/2 \pm k$ are in the NS/R sectors.

On the other hand, it is possible to compute the momentum space wave functions from the eigenstate of the Schrödinger equation (16) as

$$\psi_{n,K}(k)_L = \frac{1}{\sqrt{L}} \sum_{x} \psi_{n,K}(x) e^{ikx}. \tag{46}$$

For the choice of purely real wave functions in space, these amplitudes are imaginary due to the oddity of the spatial wave functions. While these are expected to match the amplitudes (45) up to a possible phase difference, the actual relation turns out to be more subtle:

$$\widetilde{\psi}_{n,K}(k)_L = \psi_{n,K}(k)_L \, e^{i\beta_n(K)_L} \begin{cases} 1, & \text{if } (K/2+k)(K/2-k) > 0, \\ -1, & \text{if } (K/2+k)(K/2-k) < 0, \\ i, & \text{if } (K/2+k)(K/2-k) = 0 \text{ and } K < 0, \\ -i, & \text{if } (K/2+k)(K/2-k) = 0 \text{ and } K > 0. \end{cases} \tag{47}$$

In this formula, $\beta_n(K)_L$ is the expected, non-physical overall phase that needs to be fixed for every state separately and depends on accidental phase choices for the eigenvectors obtained using numerical diagonalisation.[4] However, there is a non-trivial momentum-dependent relative phase factor in (45), which can be attributed to a relative phase between the Bogolyubov

---

[4]While the knowledge of these phases is not essential in our considerations, they must be taken into account if one wishes to reproduce the time evolutions displayed in Figs. 1a and 2a in terms of the overlaps.

fermions $\eta_q$ definition on the chain and those in the quantum mechanical description. Denoting the latter by $a_q$, the matching rules (47) correspond to the relation

$$\eta_k = \sigma_k a_k, \quad \sigma_k = \begin{cases} +1, & \text{if } k > 0, \\ -1, & \text{if } k < 0, \\ -i, & \text{if } k = 0. \end{cases} \tag{48}$$

At present, we do not have a formal derivation of this rule. However, robust evidence is provided by examining the wave function matching numerically, as shown in Appendix B. Here, we only note that these rules work equally well for meson and bubble wave functions and all possible momenta and species numbers.

## 4.3 Meson overlaps

When the post-quench dynamics is confining, the expansion of the initial state in post-quench eigenstates must be performed in terms of multi-meson states. While this seems rather complex, it is greatly simplified by the validity of the two-fermion approximation, which allows for several simplifications:

- The state (37), constructed neglecting the longitudinal field, can be considered an excellent approximation of the actual initial state. In particular, the exact initial state also lies predominantly in the two-fermion subspace. As a result, the expansion is dominated by single-meson states.

- The single-meson state is very well approximated by the amplitudes $\widetilde{\psi}_{n,K}$, constructed from the Schrödinger wave functions $\psi_{n,K}$ using the correspondence (47).

These features allow a semi-analytic calculation where the initial state is approximated by the analytic expression derived in Subsection 4.1, while the meson states are computed in the two-fermion approximation by solving the Schrödinger equation (16). The latter, while non-analytic, can be performed to any desired accuracy; therefore, the final overlaps can be considered free of numerical errors. Additionally, the method allows for the overlaps to be computed directly for a chain of any length, making the results applicable in the thermodynamic limit of an infinite system. In practice, we computed the overlaps for a chain of length $L = 100$, which effectively discretises the momentum integral corresponding to the infinite volume limit.

We also note here that the approximation of the initial state by neglecting the longitudinal field can be further validated by investigating the time evolution after the confining quench with the initial state exactly corresponding to the $h_l = 0$ scenario

$$|\Psi(0)\rangle = \sigma^z_{L/2} |+\rangle_{h_t, h_l = 0}, \tag{49}$$

which has a small global component. An example of such a quench is presented in Fig. 15 a) of Appendix C, for the parameters $h_t = 0.25$ and $h_l = 0.2$. To the naked eye, there is no obvious difference between the time evolutions of the longitudinal magnetisation shown in Fig. 15 a) and of the purely local quench shown in the second panel of Fig. 1b. Indeed, the deviation turns out to be suppressed by around 3 orders of magnitude compared to the signal from the localised component of the purely local quench.

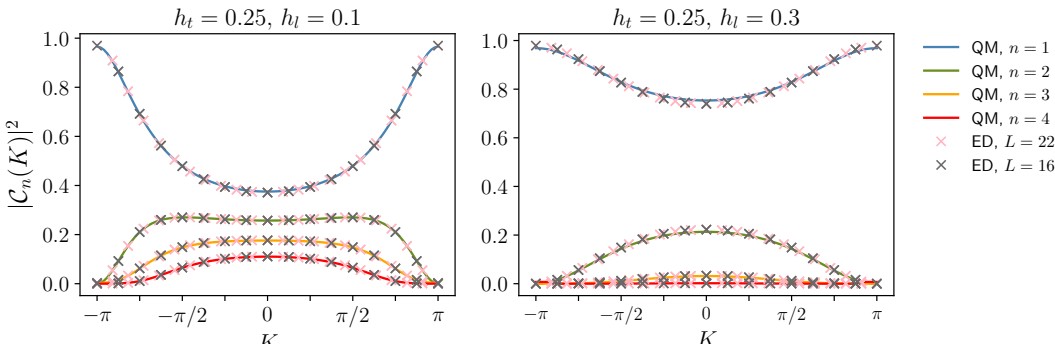

Figure 9: Comparison of the meson overlaps with the (flipped) true vacuum state $\sigma^z_{L/2}|+\rangle_{h_t,h_l=0}$ calculated by applying exact diagonalisation (shown by crosses), and the same calculated via the semi-analytic method given by Eq. (52) (shown by continuous curves) for the cases $h_t = 0.25$, $h_l = 0.1$ and $h_l = 0.3$.

Using the results of Subsection 4.1, our approximation for the initial state of the local quench can be written in the following form

$$|\Psi(0)\rangle \approx \mathcal{A}|0\rangle_{\text{NS}} + \mathcal{B}|0\rangle_{\text{R}} + \frac{1}{L}\sum_{k_1,k_2 \in \text{NS}} \mathcal{K}(k_1,k_2)_L |k_1,k_2\rangle_{\text{NS}} + \frac{1}{L}\sum_{\substack{k_1,k_2 \in \text{R} \\ k_1,k_2 \neq 0}} \mathcal{K}(k_1,k_2)_L |k_1,k_2\rangle_{\text{R}}$$

$$+ \frac{1}{L}\sum_{\substack{k_2 \in \text{R} \\ k_2 \neq 0}} \mathcal{K}(0,k_2)_L |0,k_2\rangle_{\text{R}}, \tag{50}$$

where $|k_1,k_2\rangle_{\text{NS/R}}$ is a short notation for the two-fermion state $\eta^\dagger_{k_1}\eta^\dagger_{k_2}|0\rangle_{\text{NS/R}}$, and $\mathcal{K}(k_1,k_2)_L$ denotes the corresponding amplitudes. This state can be expanded in terms of the meson states $|M_n(K)\rangle$ as

$$|\Psi(0)\rangle = \frac{1}{\sqrt{L}}\sum_{n,K} \mathcal{C}_n(K)|M_n(K)\rangle, \tag{51}$$

where[5]

$$\mathcal{C}_n(K) = \frac{1}{\sqrt{L}}\sum_{\substack{k \in \text{NS/R} \\ k \neq 0, \pm K/2}} \sigma_{K/2-k}\sigma_{K/2+k}\psi^*_{n,K}(k)_L \mathcal{K}(K/2-k, K/2+k)_L$$

$$+ \frac{1}{\sqrt{L}}\sigma_0\sigma_K\psi^*_{n,K}(K/2)_L \mathcal{K}(0,K)_L, \tag{52}$$

are the meson overlaps which satisfy

$$|\mathcal{C}_n(K)|^2 = |\mathcal{C}_n(-K)|^2, \tag{53}$$

due to symmetry under spatial reflection. The resulting overlaps are compared to exact diagonalisation for chain lengths $L = 16$ and $22$ in Fig. 9, showing excellent agreement. Note that for the larger values of the longitudinal field $h_l$, small deviations are visible from exact diagonalisation, which correspond to the limitations of the linear potential approximation of meson states discussed in Subsection 2.1, and the limitations of the approximation of the initial state by setting $h_l = 0$.

---

[5]Note that in Eqn. (52) the sum is not restricted to avoid double summation.

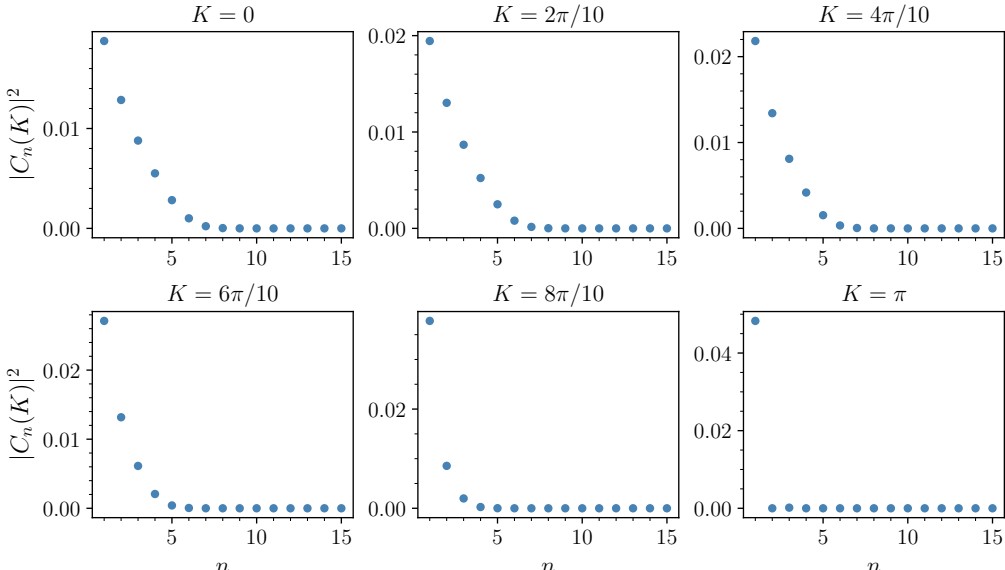

Figure 10: Overlaps of meson states $|M_n(K)\rangle$ with the state $\hat{\sigma}^z_{L/2}|+\rangle_{h_t,h_l=0}$ (given by (37)), calculated via the semi-analytic method given by Eq. (52), as a function of $n$ at different values of momentum $K$.

These results show that the post-quench evolution of confining quenches is dominated by single-meson states, mostly by the lowest-lying meson $M_1$. The state can be rewritten in the suggestive form

$$|\Psi(0)\rangle = \frac{1}{\sqrt{L}}\bigg\{\sum_n \mathcal{C}_n(0)|M_n(0)\rangle + \sum_n \mathcal{C}_n(\pi)|M_n(\pi)\rangle$$
$$+ \sum_n \sum_{0<K<\pi}\Big[\mathcal{C}_n(K)|M_n(K)\rangle + \mathcal{C}_n(-K)|M_n(-K)\rangle\Big]\bigg\}. \qquad (54)$$

The first two terms describe zero-momentum mesons building up the localised oscillations visible in Fig. 1. At the same time, the sum contains single mesons in a superposition of a left-moving and a right-moving state with the same velocity contributing with equal weights, corresponding to "Schrödinger kitten" states. These states are present because the system's initial state is not translationally invariant, unlike after global quenches. Due to the symmetry of the local quenches applied to the system, the expectation value of the total momentum is still required to be zero, which is ensured by the relation Eq. (53) that is indeed satisfied by the calculated overlaps. The emergence of the prominent light cone is explained by these "Schrödinger kitten" states escaping confinement since the initial state is decisively dominated by single meson states and the calculated overlaps of the "Schrödinger kitten" states are of the same order of magnitude as the overlaps of the zero-momentum meson states building up the localised oscillations as shown in Fig. 9.

We note that meson overlaps are suppressed with increasing index $n$ due to the large-$n$ momentum space wave function (28) oscillating rapidly with $k$, as shown in detail in Fig. 10.

## 5 Escaping fronts in anti-confining local quenches

The situation is similar for anti-confining quenches, with bubbles replacing the mesons. Following the same reasoning as in the previous subsection, the initial state can be written in the

form

$$|\Psi(0)\rangle \approx \mathcal{A}|0\rangle_{\text{NS}} - \mathcal{B}|0\rangle_{\text{R}} + \frac{1}{L}\sum_{k_1,k_2 \in \text{NS}} \mathcal{K}(k_1,k_2)_L |k_1,k_2\rangle_{\text{NS}} - \frac{1}{L}\sum_{\substack{k_1,k_2 \in \text{R} \\ k_1,k_2 \neq 0}} \mathcal{K}(k_1,k_2)_L |k_1,k_2\rangle_{\text{R}}$$

$$- \frac{1}{L}\sum_{\substack{k_2 \in \text{R} \\ k_2 \neq 0}} \mathcal{K}(0,k_2)_L |0,k_2\rangle_{\text{R}} , \tag{55}$$

where the relative sign between the NS and R components corresponds to the state built by a spin-flip performed in the false vacuum as discussed at the end of Subsection 4.1. We note that the bubble wave functions in momentum space contain a similar relative sign between the NS and R components, again reflecting that these excitations are built upon the false vacuum, and these two signs eventually cancel when computing the overlaps

$$\overline{\mathcal{C}}_n(K) = \frac{1}{\sqrt{L}}\sum_{k \in \text{NS}} \sigma_{K/2-k}\sigma_{K/2+k}\overline{\psi}^*_{n,K}(k)_L \mathcal{K}(K/2-k, K/2+k)_L$$

$$+ \frac{1}{\sqrt{L}}\sum_{\substack{k \in \text{R} \\ k \neq 0, \pm K/2}} \sigma_{K/2-k}\sigma_{K/2+k}\overline{\psi}^*_{n,K}(k)_L \mathcal{K}(K/2-k, K/2+k)_L$$

$$+ \frac{1}{\sqrt{L}}\sigma_0\sigma_K\overline{\psi}^*_{n,K}(K/2)_L , \mathcal{K}(0,K)_L , \tag{56}$$

which gives the expansion of the initial state in the bubble states of the post-quench Hamiltonian:

$$|\overline{\Psi}(0)\rangle = \frac{1}{\sqrt{L}}\sum_{n,K}\overline{\mathcal{C}}_n(K)|B_n(K)\rangle . \tag{57}$$

Once again, the bubble overlaps satisfy the symmetry property

$$|\overline{\mathcal{C}}_n(K)|^2 = |\overline{\mathcal{C}}_n(-K)|^2 , \tag{58}$$

and are in excellent agreement with the exact diagonalisation results, as shown in Fig. 11. We only display the results for $h_l = 0.1$; increasing the field value pushes the false vacuum energy higher, making it harder to identify the bubble states by exact diagonalisation due to the levels overlapping with a dense spectrum of two-meson states. Note that the qualitative form of the overlap functions is similar to that of the confining quench. Additionally, the overlaps are suppressed with growing species index $n$, as illustrated in Fig. 12, similarly to the behaviour of meson overlaps discussed in Subsection 4.3. This is important because the bubble spectrum is unbounded from below, which prevents a direct application of the energy arguments applied to anti-confining quenches at the beginning of Section 4. This is especially true for supercritical bubbles (i.e., larger the size $n_*$ given in Eqn. (34)), which have negative energy compared to the false vacuum initial state; however, the behaviour of the overlaps strongly suppresses their contribution. As a result, overlaps of supercritical bubbles are many orders of magnitude smaller than 1.

These results and considerations lead to the following conclusions:

- The localised oscillations correspond to bubbles with momenta $K = 0$ or $\pi$.

- The escaping fronts are formed by single-bubble states in superpositions of a left-moving and a right-moving component of exactly opposite velocities with the same weight.

The latter conclusion is further supported by the fact that the maximum velocity of the smallest bubble $B_1$ matches the escaping fronts well, as shown in Fig. 2a.

To sum up, for the anti-confining quenches, dynamical confinement is replaced by Wannier-Stark localisation but again, it is the superpositions of left and right-moving states that escape it, with the only difference that these are bubble states instead of mesonic ones.

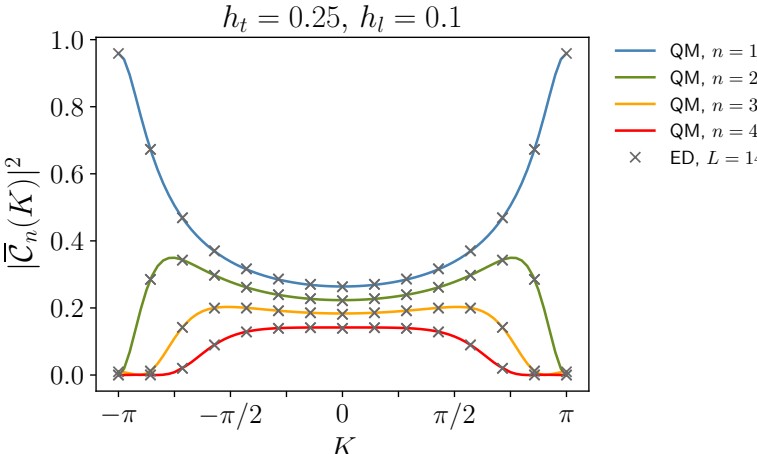

Figure 11: Comparison of the bubble overlaps with the (flipped) false vacuum state of the pure transverse system $\sigma^z_{L/2}\,|-\rangle_{h_t,h_l=0}$ calculated from exact diagonalisation (crosses) with the same calculated via the semi-analytic method given by Eq. (56) at the parameter values $h_t = 0.25$ and $h_l = 0.1$ (continuous curves).

## 6 Conclusions and Outlook

We considered local quenches induced by a single spin flip on the Ising spin chain in the ferromagnetic regime. In the presence of longitudinal fields, the chain dynamics is confining, and the quasi-particles over the ground state (the true vacuum) are bound states of kink pairs, which are analogous to the mesons of strong interactions. In global quenches, the confining force strongly suppresses the propagation of quasi-particles [3].

In contrast with global quenches, our simulations show that after local quenches are obtained by flipping a single spin in the ground state, the magnetisation has a significant propagating component besides the localised one. To determine the origin of these fronts, we computed the overlaps of meson states with the initial state using a semi-analytic approach, which works for any chain length, including the thermodynamic limit. The conclusion is that the light-cone fronts originate from states containing a single meson excitation in superpositions of left and right-moving mesons (mainly the lightest one) with equal weights, which is in contrast to global quenches where quasi-particle pairs trace out the light cones and are strongly suppressed by confinement. The interpretation of the global quench scenario is closer to our classical intuition since the two branches of the light cone are traced out by pairs of quasi-particles with opposite momentum. However, in local quenches, it is one-particle states in the superposition of left and right moving momenta with equal amplitudes that are responsible for the presence of the emergent light cone.

We also considered anti-confining quenches initiated by a single spin-flip in the false vacuum. In global quenches, the light cone spreading of correlations is again suppressed; however, this time by the mechanism of Wannier-Stark localisation [26] based on Bloch oscillations, which originate from the periodicity of the lattice dispersion relation in momentum space, and also result in the suppression of the decay of the false vacuum [38]. In the local quenches considered here, we once again observed a significant propagating component of magnetisation besides the localised one. This time, the excitations responsible for the fronts are bubbles of the true vacuum nucleating from the false vacuum, again in superpositions of left and right-moving bubbles (dominated by the one with the smallest spatial extension) with equal weights. Interestingly, in infinite volume, the spectrum of these bubbles is bounded from above while unbounded from below. In contrast, in a finite volume, their spectrum seamlessly blends into

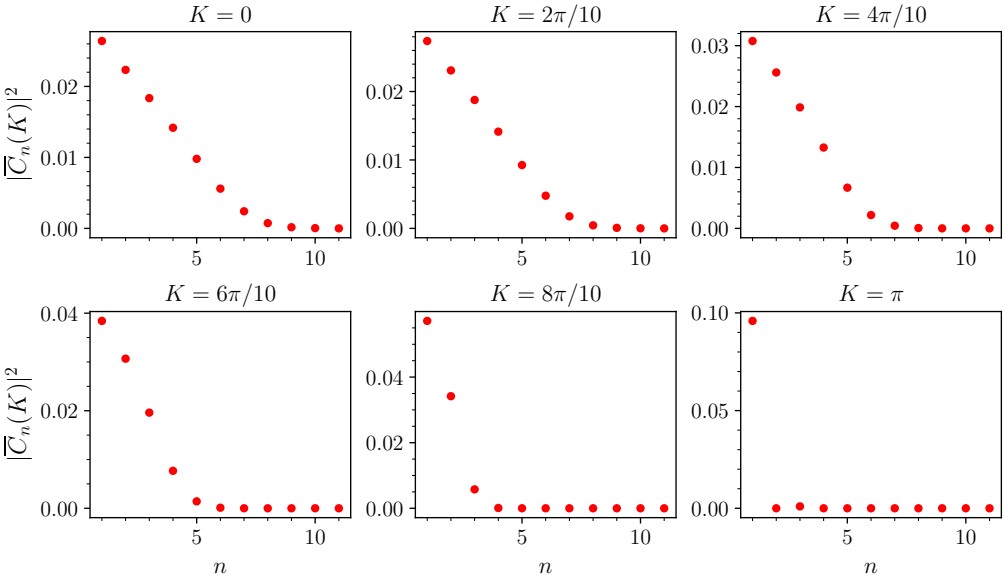

Figure 12: Overlaps of bubble states $|B_n(K)\rangle$ with the with the state $\sigma^z_{L/2}|-\rangle_{h_t,h_l=0}$ (given by (42)), calculated via the semi-analytic method given by Eq. (56), as a function of $n$ at different values of momentum $K$.

that of the mesons since both large bubbles and large mesons correspond to collisionless kink pairs separated by distances much larger than the Wannier-Stark localisation length.

Beyond the spin-flip scenario considered here, we note that an interesting local quench can be obtained by starting from a domain wall initial state, for which all light cone propagation is suppressed similarly to global quenches [26]. We included our simulation of this situation in Appendix C; the absence of propagating fronts is explained by combining arguments from energy conservation with the behaviour of meson and bubble overlaps, specifically their suppression with the spatial size of the corresponding excitation established in Sections 4 and 5.

It is also interesting to perform a quantum simulation of these phenomena on a quantum computer or in a suitably designed ultra-cold atomic experiment. While confining local quenches have previously been considered on a quantum computer [31], we are unaware of similar results for anti-confining quenches. Another interesting direction is to consider these phenomena in the framework of the quantum 3-state Potts spin chain in its confining phase, where the spectrum contains baryonic excitations besides the mesonic ones [60–63], strengthening the analogy with the strong interactions and enlarging the scope of potential physical phenomena.

# Acknowledgments

We are grateful to M.A. Werner for sharing his extensive knowledge of MPS methods and helping us develop our simulations. We also thank M. Kormos and M. Lencsés for discussions, and are grateful to A. Bastianello for useful comments on the initial version of the manuscript.

**Funding information** This work was supported by the National Research, Development and Innovation Office (NKFIH) through the OTKA Grants SNN 139581 and K 138606. AK was also partially supported by the Doctoral Excellence Fellowship Programme (DCEP), funded by the National Research Development and Innovation Fund of the Ministry of Culture and Innovation

and the Budapest University of Technology and Economics, under a grant agreement with the National Research, Development and Innovation Office. GT was partially supported by the Quantum Information National Laboratory of Hungary (Grant No. 2022-2.1.1-NL-2022-00004).

# A   Exact solution of the purely transverse Ising chain

Here, we briefly recall the analytic solution of the TFIM [64] with length $L$ and periodic boundary conditions described by the Hamiltonian

$$H = -J \sum_{j=1}^{L} \left( \hat{\sigma}_j^x \hat{\sigma}_{j+1}^x + h_t \hat{\sigma}_j^z \right), \tag{A.1}$$

with $h_t$ denoting the transverse magnetic field. The spin ladder operators

$$\hat{\sigma}_j^{\pm} = \frac{1}{2} \left( \hat{\sigma}_j^x \pm \hat{\sigma}_j^y \right), \tag{A.2}$$

can be expressed in terms of fermionic operators $c_j$ and $c_j^{\dagger}$ satisfying

$$\{c_j, c_l^{\dagger}\} = \delta_{jl}, \qquad \{c_j, c_l\} = \{c_j^{\dagger}, c_l^{\dagger}\} = 0, \tag{A.3}$$

via the Jordan-Wigner transformation [65]

$$c_j = W_j \hat{\sigma}_j^{\dagger}, \qquad c_j^{\dagger} = \hat{\sigma}_j^- W_j, \tag{A.4}$$

using the string operators

$$W_j = \prod_{l=1}^{j-1} \hat{\sigma}_l^z, \qquad W_1 = \mathbb{1}. \tag{A.5}$$

The Hamiltonian (A.1) can be rewritten as

$$H = -J \sum_{j=1}^{L-1} \left( c_j^{\dagger} - c_j \right) \left( c_{j+1} + c_{j+1}^{\dagger} \right) + J h_t \sum_{j=1}^{L} \left( c_j^{\dagger} c_j - c_j c_j^{\dagger} \right) - J e^{i\pi \hat{N}} \left( c_L - c_L^{\dagger} \right) \left( c_1 + c_1^{\dagger} \right), \tag{A.6}$$

where $\hat{N}$ is the fermion number operator

$$\hat{N} = \sum_{j=1}^{L} c_j^{\dagger} c_j. $$

For simplicity, we now restrict the length $L$ of the chain to be even. Since $H$ and $\hat{N}$ commute, the Hamiltonian is block diagonal in the subspaces of even/odd fermion number. The last term of (A.6) can be absorbed into the first sum as

$$H = -J \sum_{j=1}^{L} \left( c_j^{\dagger} - c_j \right) \left( c_{j+1} + c_{j+1}^{\dagger} \right) + J h_t \sum_{j=1}^{L} \left( c_j^{\dagger} c_j - c_j c_j^{\dagger} \right), \tag{A.7}$$

provided that the fermions satisfy the boundary conditions

$$\begin{cases} N \text{ even:} & c_{L+1} = -c_1, \\ N \text{ odd:} & c_{L+1} = c_1. \end{cases} \tag{A.8}$$

These correspond to two subspaces of the Hilbert space called the Neveu-Schwarz (NS) / Ramond (R) sectors. The fermions are transformed into Fourier space as

$$b_q = \frac{1}{\sqrt{L}} \sum_{j=1}^{L} c_j e^{iqj}, \quad c_j = \frac{1}{\sqrt{L}} \sum_q b_q e^{-iqj}, \tag{A.9}$$

where the allowed values of the momenta for the NS sector are

$$q_n^{\text{NS}} := k_n = \frac{2\pi(n+1/2)}{L}, \quad n = -\frac{L}{2}, \dots \frac{L}{2} - 1,$$

while for the R sector

$$q_n^{\text{R}} := p_n = \frac{2\pi n}{L}, \quad n = -\frac{L}{2}, \dots \frac{L}{2} - 1. \tag{A.10}$$

The Hamiltonian then takes the form

$$H = -J \sum_q \left( 2 \left( \cos q - h_t \right) b_q^\dagger b_q + h_t + e^{-iq} b_q^\dagger b_{-q}^\dagger - e^{iq} b_q b_{-q} \right), \tag{A.11}$$

and can be diagonalised using a Bogolyubov transformation in both sectors, which has the following general form for modes with nonzero momenta

$$b_q = u_q \eta_q + i v_q \eta_{-q}^\dagger, \qquad b_{-q}^\dagger = u_q \eta_{-q}^\dagger + i v_q \eta_q, \quad q \neq 0, \tag{A.12}$$

where

$$u_q = \frac{h_t - \cos q + \sqrt{1 + h_t^2 - 2h_t \cos q}}{\left( \sin^2 q + \left( h_t - \cos q + \sqrt{1 + h_t^2 - 2h_t \cos q} \right)^2 \right)^{1/2}},$$

$$v_q = \frac{-\sin q}{\left( \sin^2 q + \left( h_t - \cos q + \sqrt{1 + h_t^2 - 2h_t \cos q} \right)^2 \right)^{1/2}}. \tag{A.13}$$

For zero momentum in the Ramond sector, the definition of the new modes depends on the phase:

$$\begin{aligned} b_0 &= \eta_0, & b_0^\dagger &= \eta_0^\dagger, & h_t > 1 & \quad \text{PM phase,} \\ b_0 &= \eta_0^\dagger, & b_0^\dagger &= \eta_0, & h_t < 1 & \quad \text{FM phase.} \end{aligned} \tag{A.14}$$

Using this convention, the Hamiltonian in both sectors takes the following form

$$H = \sum_{q \in \text{NS/R}} \varepsilon(q) \left( \eta_q^\dagger \eta_q - \frac{1}{2} \right),$$

with the dispersion relation

$$\varepsilon_q = 2J \sqrt{1 + h_t^2 - 2h_t \cos q}. \tag{A.15}$$

In both sectors, it is possible to define a Fock ground state by

$$\eta_q |0\rangle_{\text{NS/R}} = 0, \qquad q \in \text{NS/R}. \tag{A.16}$$

However, the spectrum is very different in the two phases. In the paramagnetic phase, the Hilbert space is generated by states containing an even number of excitations over the $|0\rangle_{\text{NS}}$,

while an odd number of excitations over the $|0\rangle_R$. Then the system has a single ground state identical to $|0\rangle_{\text{NS}}$, while the lowest energy state in the Ramond sector is the one-particle state

$$\eta_0^\dagger |0\rangle_R \,, \tag{A.17}$$

which has a finite gap $2J(h-1)$ over the ground state in the thermodynamic limit. The NS/R sectors contain states with an even/odd number of excitations over the corresponding vacuum states, respectively, so the vacuum vector $|0\rangle_R$ is not part of the spectrum.

In the ferromagnetic phase, both sectors contain states with an even number of excitations over the corresponding vacuum states. The energy difference between the two ground states is

$$E_R - E_{\text{NS}} = -\frac{1}{2}\sum_{q\in R}\varepsilon(q) + \frac{1}{2}\sum_{q\in \text{NS}}\varepsilon(q)\,, \tag{A.18}$$

which vanishes exponentially in the thermodynamic limit $L \to \infty$. The system manifests a spontaneous symmetry breaking with the combinations

$$|\pm\rangle = \frac{1}{\sqrt{2}}\left(|0\rangle_{\text{NS}} \pm |0\rangle_R\right)\,, \tag{A.19}$$

corresponding to the two ground states (assuming an appropriate choice of the relative phase of the states $|0\rangle_{\text{NS/R}}$). In a finite system, these states are connected by tunnelling, which is exponentially suppressed in $L$. The expectation value of the order parameter is

$$\langle\pm|\sigma_j^x|\pm\rangle = \pm\left(1 - h_t^2\right)^{1/8} + \dots\,, \tag{A.20}$$

where the ellipsis indicates a correction vanishing exponentially in the system size.

## B Numerical results for meson and bubble wave functions

Here, we compare explicitly the wave functions computed in the two-fermion approximation to those obtained from exact diagonalisation in momentum space. The real-space wave functions obtained from Eqs. (16,30) are transformed to Fourier space using

$$\psi(k) = \frac{1}{\sqrt{L}}\sum_x e^{ikx}\psi(x)\,, \tag{B.1}$$

where the normalisation is set to match the one on a periodic chain of length $L$ (the sum over $x$ can be considered to run from $-\infty$ to $\infty$ since the wave functions vanish rapidly for large $x$ and it is easy to obtain them in a volume much larger than their localisation). This is compared to the wave functions obtained from exact diagonalisation

$$\widetilde{\psi}_{n,K}(k)_L = {}_{\text{NS/R}}\langle 0|\eta_{K/2+k}\eta_{K/2-k}|M_n(K)\rangle\,,$$
$$\widetilde{\overline{\psi}}_{n,K}(k)_L = {}_{\text{NS/R}}\langle 0|\eta_{K/2+k}\eta_{K/2-k}|B_n(K)\rangle\,, \tag{B.2}$$

modified by the (inverse of the) phase factors specified in (47). This results in an excellent match, as illustrated in Fig. 13 for mesons and in Fig. 14 for bubbles, even on a chain of moderate length.

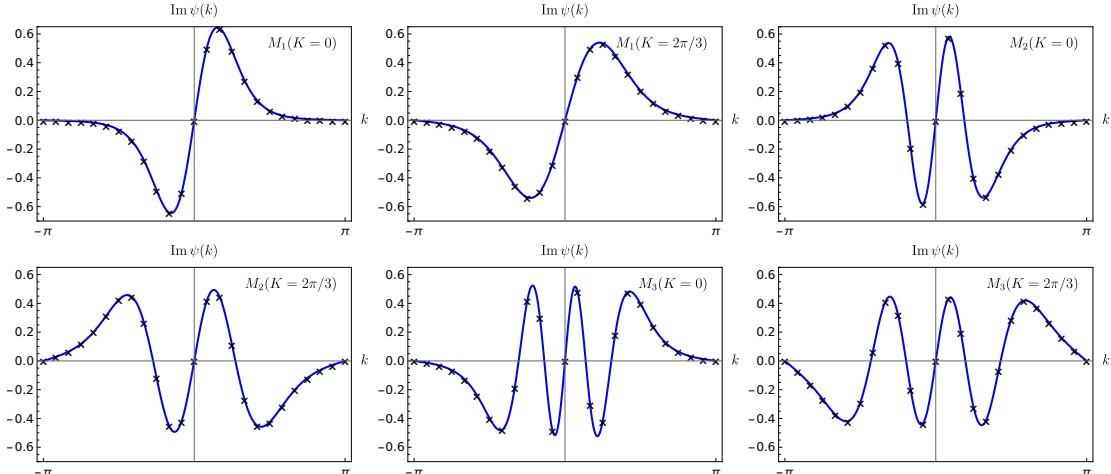

Figure 13: Meson wave functions at $h_t = 0.25$, $h_l = 0.1$ compared to exact diagonalisation with $L = 12$.

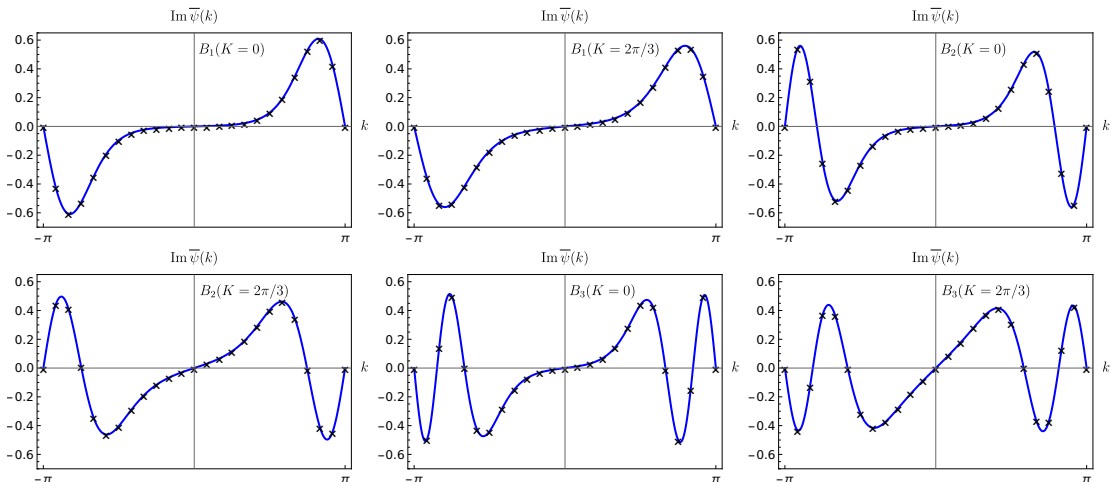

Figure 14: Bubble wave functions at $h_t = 0.25$, $h_l = 0.1$ compared to exact diagonalisation with $L = 12$.

## C  Time evolution after other local quenches

To further support the main text's arguments and gain better insight, we compare other quenches to the purely local confining quench with the initial state $|\Psi(0)\rangle = \sigma^z_{L/2} |+\rangle_{h_t,h_l}$.

The first example shows that neglecting the global quench inherent in approximating the integrability breaking initial states ($h_l \neq 0$) by the integrable ones ($h_l = 0$) as done in Section 4 produces a negligible effect and does not change the phenomenology. This approximation is supported by examining confining quenches starting from the integrable initial state

$$|\Psi(0)\rangle = \sigma^z_{L/2} |+\rangle_{h_t, h_l=0} \,, \tag{C.1}$$

presented in Fig. 15 a) for $h_t = 0.25$ and $h_l = 0.2$. This quench differs from the one considered in Section 2 by including a global component corresponding to the change from $h_l = 0$ to $h_l = 0.2$. However, the difference between the time evolution shown in Fig. 1 for $h_l = 0.2$ and in Fig. 15 a) is not visible to the naked eye; it turns out to be suppressed by around 3 orders of magnitude compared to the signal from the local part of the quench.

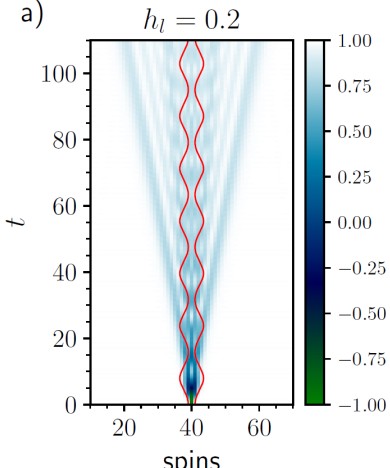
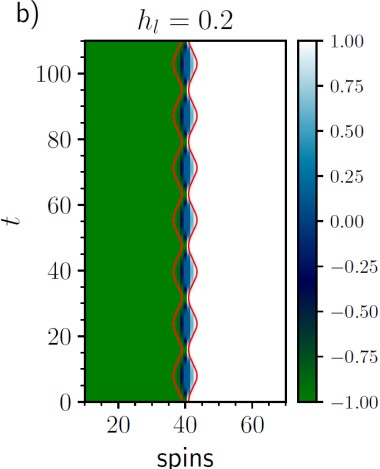

Figure 15: a) Time evolution of the longitudinal magnetisation $\langle \hat{\sigma}_j^x \rangle$ starting from the initial state $|\Psi(0)\rangle = \sigma_{L/2}^z |+\rangle_{h_t, h_l=0}$ for the values $h_t = 0.25$ and $h_l = 0.2$. The red curves indicate the corresponding quasi-classical trajectories of the quasi-particles with momenta $k_0 = \pm\pi$. b) Time evolution of the longitudinal magnetisation $\langle \hat{\sigma}_j^x \rangle$ starting from the initial state $|\Psi(0)\rangle = \prod_{j=1}^{L/2} \sigma_j^z |+\rangle_{h_t, h_l}$ (domain wall state) for the values $h_t = 0.25$ and $h_l = 0.2$. The red curves on the confining side with positive magnetisation (right side of the system) indicate the corresponding quasi-classical trajectories of the quasi-particles with momenta $k_0 = \pm\pi$, while on the anti-confining side with negative magnetisation (left side of the system) the red curves indicate the trajectories corresponding to the momentum $k_0 = 0$.

Another interesting quench protocol can be realised by choosing the initial state as

$$|\Psi(0)\rangle = \prod_{j=1}^{L/2} \sigma_j^z |+\rangle_{h_t, h_l} \, , \tag{C.2}$$

which is a "domain wall" state. In this scenario, the system segment characterised by positive magnetisation corresponds to a confining quench, while the other half of the system is an anti-confining quench. A simpler example of such a time evolution, in which the two halves were initialised with all spins up/down, respectively, was considered in [22, 26]. Similarly to their results, the time evolution of the longitudinal magnetisation only exhibits a local component which does not spread in time as shown in Fig. 15 b) for $h_t = 0.25$ and $h_l = 0.2$.

The trajectories shown in Fig. 15 b) were obtained from (12). For the confining side, the localised part is expected to be bounded by the quasi-classical trajectory of a kink with the maximum possible initial momentum $k_0 = \pi$. In contrast, on the anti-confining side, the boundary is expected to be traced out by a kink with initial momentum $k_0 = 0$ function, as shown in Fig. 15.

The suppression of escaping fronts in the domain wall quench is in marked contrast to the quenches initialised by local spin-flips considered in Section 2. This can be understood on the confining side of the quench by reconsidering the energy balance argument presented at the beginning of Section 4. The energy locally fed into the system by the presence of the domain wall is roughly smaller by a factor of $1/2$ due to the presence of a single boundary, instead of the two which occur at a spin-flip and is therefore insufficient for the creation of even the lowest lying meson state. However, this argument is insufficient as an explanation because, on the anti-confining side, the bubble spectrum is unbounded from below, seemingly allowing for the

unlimited creation of bubbles. Nevertheless, note that bubbles with small spatial extensions (i.e., species index) have positive energy and only supercritical bubbles with sizes larger than $n_*$ in Eqn. (34) can turn the energy balance in a favourable direction. As discussed in Section 5, the size of these bubbles is so large that they have extremely small overlaps with the initial state. Additionally, such supercritical bubbles are collisionless and therefore stationary, as discussed in Section 3.

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
