# Peer review of "Escaping fronts in local quenches of a confining spin chain"

_SciPost Physics, doi:SciPost Phys. 16, 138 (2024)_

## Round 2 · Referee Report · Anonymous · 2024-3-1

Strengths

- explores novel features of confining queches
- careful numerical and analytical results
- clean and readable presentation

Weaknesses

- misleading interpretation in terms of cat states

Report

The authors study quench dynamics in the confining Ising spin chain, i.e. in the presence of both transverse and longitudinal field components. Previous studies of this model have pointed out that, in the ferromagnetic phase, a small longitudinal field component introduces an attractive force between the kink-excitations of the non-interacting (purely transverse) Ising chain. Then the low-lying spectrum can effectively be described in the two-kink approximation, and the corresponding quasiparticles are mesons. These can be understood as bound states of two kinks, and the confining behaviour of mesons has first been observed in global quenches as an almost complete suppression of the light-cone signal propagation.

In the present contribution, the authors consider a purely local quench and demonstrate that, in sharp contrast to the global case, the magnetization front shows a clear light-cone behavior. This is observed in both the confining quench, as well as in the anti-confining one, which corresponds to starting the quench from the false vacuum (i.e. antiparralel to the longitudinal field). The goal of the manuscript is to understand, how the escaping of confinement could be understood in terms of the underlying quasiparticles, namely meson or bubble excitations, respectively.

First, it is shown that the two-kink approximation works perfectly, by comparing the spectrum of the effective Hamiltonian to that of the Ising chain, and observing perfect match already at small chain sizes. It is shown that the higher bands of mesons correspond to an increasing separation between the kinks. Moreover, it is pointed out that the bubble and meson energy levels can be matched in the middle of the spectrum. The analysis then continues with the study of the overlaps between the meson/bubble states and the (non-interacting) ground state. Again, the analytical calculations are in perfect agreement with the values obtained from exact diagonalization. In particular, one finds that the quasiparticle weights for large $n$ are heavily suppressed, and the observed light-cones correspond to the largest group velocities of the family $n=1$.

In my opinion, the manuscript is nicely written and the topic is interesting. The results are sound and illustrated by nice figures. However, there is a serious issue regarding the interpretation of the results in terms of Schrödinger cats, which is highly misleading.

Requested changes

1) According to standard convention, a Schrödinger cat is a quantum state
composed by the superposition of two, macroscopically different states
(e.g. the ground states in the NS/R sectors are such cat states).
In fact, none of the above criteria applies to the result of the authors.
Clearly, (4.16) is a superposition of a huge number of different meson
states with different momenta $K$. The fact that it can be written by
equal weights for $\pm K$ is a trivial consequence of reflection symmetry,
and it does not add new insight. Furthermore, the corresponding states
are also not macroscopically different. Indeed, as the relevant meson
states are the ones with small $n$, these just correspond to localized
excitations, where the kinks are close to each other, as discussed in sec. 3.
Instead, a cat state would be e.g. the superposition of the configurations
in Fig. 3.6., i.e. a meson and a bubble, which is clearly not the case.

2) I believe that the correct interpretation of the results is much simpler,
and does not require any mystification in terms of cat states.
For a global quench the ground state is excited homogeneously, which
corresponds to exciting only QPs with total momentum zero, i.e. $K=0,\pi$.
These are immobile excitations with zero group velocity, as is clear
from the dispersion in Fig. 3.1. In contrast, for a quench that is local
in space, one expects that QPs are excited in a broad momentum range $K$,
as confirmed perfectly by the results. These mesons/bubbles are, however,
mobile with finite group velocities and they will trace out a light cone.

In my opinion, the authors should reformulate their interpretation and
remove the misleading reference to Schrödinger cats (including the
title of the paper), unless they have some very strong argument
I might have overlooked.

3) It would be useful to move Figs 2.3 and 2.4 together (like 2.1 and 2.2)
to allow for a better comparison.

4) In the right of Fig. 3.2, the eigenvector with n=2 could be multiplied
by (-1), to facilitate comparison of left and right panels.

5) Why is a different notation introduced in (4.1)? It seems like this is
identical to $|+\rangle_{h_t,h_l=0}$

6) Isn't the explanation of (4.13) somehow related to the fact, that the
Fourier transform in (4.11) is defined w.r.t. the relative momentum,
whereas for Bogolyubov fermions it is defined by single-kink momenta?

  • validity: good
  • significance: good
  • originality: high
  • clarity: high
  • formatting: excellent
  • grammar: excellent

Author:  Gabor Takacs  on 2024-03-06  [id 4343]

(in reply to Report 1 on 2024-03-01)

We thank the referee for the careful reading and the useful comments on the manuscript. Here we only react to the one about the interpretation of the results.

Indeed, we realise that the interpretation as a Schrödinger cat is not well-explained in the paper. There are two aspects that we definitely should have explained better:

  • The somewhat counter-intuitive aspect is that while we have a left and a right moving light cone, these are traced out by a single-particle state. In global quenches, the light cone is generated by two separate particles going left and right. Here, the quench is dominated by single-particle states, which appear in equally weighted superpositions of a left-moving and a right-moving component. Hence the (playful) reference to Schrödinger cats. In this context, we also plan to include more explanation concerning the main difference between global and local quenches, which relates to the presence resp. absence of translation invariance.

  • The referee is also right that these are microscopic superpositions, so they might be better called Schrödinger kittens.

We shall make appropriate changes to the text and the title once the refereeing round is over, implementing also the other requested changes and clarifications.

---

## Round 2 · Referee Report · Anonymous · 2024-3-11

Strengths

1) Interesting out-of-equilibrium protocol.
2) Accurate analytical and numerical study.
3) Very clearly written paper.

Weaknesses

no major weaknesses except for the misleading interpretation pointed out by the first referee.

Report

The paper explores the dynamics of local quenches initiated by a single spin flip in the confined quantum Ising spin chain, focusing on scenarios where the initial states are either in the true or false vacuum states within the ferromagnetic regime.

The authors employ a combination of analytical techniques and numerical methods to study the post quench dynamics ensuing from a local spin flip in the ground state of the Ising chain with both transverse and longitudinal magnetic fields. In global quenches, a strong suppression of light cone spreading is observed. This is explained in terms of confined meson-like excitations. In contrast, the authors find that light-cone spreading persists after a local quench.
For quenches initiated from the true vacuum state, the propagating signal is described as comprising Schroedinger cats of left and right-moving mesons, which escape confinement. Conversely, quenches initiated from the false vacuum state yield a propagating signal composed of Schrödinger cats of left and right-moving bubbles, escaping Wannier-Stark localization.

The authors explain this interesting behaviour by calculating the overlaps between the initial states and the propagating excitations.

The result of the paper is interesting and the material is well presented.
My main criticism is that it is not very clear how crucial is the Schroedinger cat (or kitten) superposition structure of the excitations to escape confinement. The question would be if bubbles or mesons can propagate if they are not in a superposition. I would appreciate if the authors could comment on that.

Apart from that I think that the paper is quite interesting and meets the acceptance criteria of Scipost.

---

## Round 3 · Referee Report · Anonymous (Referee 1) · 2024-5-2

Report

The authors have followed the recommendation and removed the misleading interpretation in terms of Schrödinger cat states. The manuscript contains very nice results and can now be accepted in its present form.

Recommendation

Publish (easily meets expectations and criteria for this Journal; among top 50%)

---

## Round 3 · Author Response

We thank both Referees for their careful reading and valuable comments on the submitted manuscript. We give our replies together with the implied changes below.

Report 1:

1&2: We followed the Referee’s recommendation and reformulated the interpretation. For this, we changed the title, abstract, conclusions, and discussions at the end of Sections 4.3 and 5.

The main modification is that now we write superpositions instead of the expression ’Schrödinger cat’. We now only refer to the superpositions of the right and left-moving mesons as ’Schrödinger kitten’ states in the discussion at the end of Section 4.3 to abbreviate the text. We do not find it misleading as the reformulated interpretation itself does not build upon this notion.

In the conclusions, we added a sentence explaining that the reason we found this aspect interesting is that contrary to classical intuition and also the quasi-particle picture of global quench time evolution, for the local quenches, we get a two-pronged light cone from single-particle states, which are superpositions of left and right moving one-particle states with equal weight.

To emphasize the main difference between the local and global quenches regarding the translational in- variance, as pointed out by the Referee, we included a short explanation at the end of Section 2.1 and in the discussion part at the end of Section 4.3.

3: We agree and therefore grouped the figures (now Figs. 2.1 and 2.2) as the referee suggested.

4: We have done the requested redefinition of the eigenvector.

5: We agree and change the notation accordingly everywhere in the main text.

6: We considered several possible explanations for the eventual phases, but we could not find any fully satisfactory explanation, so we prefer to leave this as an observed fact for now.

Report 2:

The questions raised by the Referee are answered by the changes listed above under item 1&2 of our response to report 1.

---

## Round 3 · List of Changes

See resubmission letter.

---

## Editorial Decision

published